# Identification of Key Parameters Inducing Microbial Modulation during Backslopped Kombucha Fermentation

**DOI:** 10.3390/foods13081181

**Published:** 2024-04-12

**Authors:** Claire Daval, Thierry Tran, François Verdier, Antoine Martin, Hervé Alexandre, Cosette Grandvalet, Raphaëlle Tourdot-Maréchal

**Affiliations:** 1Institut Agro, Université Bourgogne Franche-Comté, Université Bourgogne, INRAE, UMR PAM 1517, 21000 Dijon, Francervalex@u-bourgogne.fr (H.A.); cosette.grandvalet@u-bourgogne.fr (C.G.); tourdot@u-bourgogne.fr (R.T.-M.); 2Biomère, 10B Rue du Nouveau Bêle, 44470 Carquefou, France

**Keywords:** kombucha, microbial ecology, reproducibility, yeasts, acetic acid bacteria

## Abstract

The aim of this study was to assess the impact of production parameters on the reproducibility of kombucha fermentation over several production cycles based on backslopping. Six conditions with varying oxygen accessibility (specific interface surface) and initial acidity (through the inoculation rate) of the cultures were carried out and compared to an original kombucha consortium and a synthetic consortium assembled from yeasts and bacteria isolated from the original culture. Output parameters monitored were microbial populations, biofilm weight, key physico-chemical parameters and metabolites. Results highlighted the existence of phases in microbial dynamics as backslopping cycles progressed. The transitions between phases occurred faster for the synthetic consortium compared to the original kombucha. This led to microbial dynamics and fermentative kinetics that were reproducible over several cycles but that could also deviate and shift abruptly to different behaviors. These changes were mainly induced by an increase in the *Saccharomyces cerevisiae* population, associated with an intensification of sucrose hydrolysis, sugar consumption and an increase in ethanol content, without any significant acceleration in the rate of acidification. The study suggests that the reproducibility of kombucha fermentations relies on high biodiversity to slow down the modulations of microbial dynamics induced by the sustained rhythm of backslopping cycles.

## 1. Introduction

As a part of human nutrition for at least 15,000 years, fermented foods were mainly associated with traditional processes for transformation and stabilization of foodstuffs. Nowadays, “foods made through desired microbial growth and enzymatic conversions of food components” have gained new interest as traditional fermented foods have transitioned from households to industrial-scale productions [1,2,3]. Fermented beverages such as kefir and kombucha led to changes in consumption behavior that now occur worldwide, which calls for modifications in regulatory frameworks [4]. This movement stemmed in recent decades from western countries and propagated worldwide thanks to globalized markets [5,6], trending values supporting well-being and self-appropriation of nutrition [7]. Moreover, an increasing amount of scientific evidence has highlighted the effective benefits of fermented food consumption, along with the unraveling of their mechanisms [8,9,10,11].

As a result, large-scale production of traditional fermented food develops and is characterized by the use of backslopped inoculation with uncontrolled microbial compositions (non-specifically selected through functional criteria), as opposed to the well-established practice of single or multi-strain inoculation [2,12]. Amplified by up-scaling production, the lack of predictability over poorly characterized microbial communities used for fermentation processes impacts quality control, thus stirring the need for tailored microbial consortia [13,14,15]. Despite being seen as one of the main obstacles in the development of traditional fermented food production, little research has been carried out on the batch-to-batch stability of fermentation processes. Studies investigated mainly sourdough and derived products [16,17,18,19,20], but also milk-based products such as Gouda cheese [21], kefir [22] and Mabisi from Zambia [23], with monitoring or endpoint duration ranging between 7 and 32 cycles (with cycle length depending on the type of fermented food). Namely, a study focusing on the impact of the inoculation method in kefir reported a depletion in yeasts associated with optimized chemical composition if backslopping was applied compared to other methods [22]. Our previous study also reported large yet reversible variations in yeast and bacteria diversity in kombucha consortia used in production contexts over 3 years [24]. Therefore, further investigation appears critical to improve the availability of traditionally fermented food on the market, especially regarding the effect of production parameters. Beyond industrial relevance, fermented food microbial systems are also seen as relevant and promising models for the study of microbial ecology and interactions [25,26,27,28].

One of them, kombucha, is a fermented, low-alcoholic sour beverage with obscure Eastern origins obtained from the transformation of sugared tea by a consortium of yeasts and bacteria, mainly acetic acid bacteria [29]. The most important biochemical transformation, the conversion of sucrose into organic acids, depends on a metabolic interplay between yeast and acetic bacteria species [14,30]. Sucrose is a carbon substrate that needs to be broken down into monosaccharides (glucose and fructose) to be consumed. This hydrolysis step is carried out efficiently thanks to yeast invertase activity, and the monosaccharides can be used for alcoholic fermentation with the production of ethanol and carbon dioxide [30,31]. Therefore, acetic acid bacteria can convert glucose and ethanol into gluconic and acetic acid, respectively, through oxidative metabolism [32]. Additional transformations include olfactive volatile compounds and modifications in polyphenol structure [15,33,34,35,36]. In parallel, a floating cellulosic biofilm is produced by acetic acid bacteria at the surface of the liquid and hosts both yeasts and bacteria within itself [15,37,38].

The aim of the study, carried out in laboratory conditions, is to assess the reproducibility of fermentation kinetics between backslopping cycles. The response of the microbial consortia was evaluated in terms of populations and activity (i.e., their modulation) to abiotic and biotic production parameters. Two different conditions of access to oxygen were tested, along with fixed or non-fixed initial acidity (abiotic parameters). At the scale of one batch, oxygen access has been confirmed as a key parameter for the acidification rate, as it is a limiting factor in the conversion of carbon substrates into organic acids [39]. In cases of insufficient oxygen access, acidification is slowed down, leading to ethanol accumulation and the depletion of sugar content [30,36]. In the present study, the two levels of access to oxygen aim at simulating industrial context (lower access) and homemade production (higher access) in relation to the Specific Interfacial Surface (SIS, defined as the liquid surface/liquid volume ratio [40]). Moreover, two kombucha cultures (original and synthetic) were implemented in the study (biotic parameters). The inoculum used for the original “O” culture came from a kombucha culture obtained from the kombucha producer Biomère (Carquefou, France). A model system was introduced and compared with the original culture to enable a more mechanistic analysis of the phenomena. Thus, for the synthetic “$” culture, the inoculum used came from a black tea infusion inoculated with strains isolated from the original kombucha culture.

## 2. Experimental Procedures

### 2.1. Culture Generation

The isolation of yeasts and bacteria from Biomère’s kombucha culture (Carquefou, France) and the culturing procedure of microorganisms were carried out based on previous studies [24,30]. Briefly, the identification and isolation of yeasts and bacteria from the original culture (“O”) were performed by culturing yeasts and bacteria on selective agar media: Wallerstein Lab agar plates from Thermo Fisher Scientific (Waltham, MA, USA) and pH 6.2 De Man Rogosa and Sharpe (MRS) agar plates from Condalab (Madrid, Spain), respectively. A set of colonies was isolated and used for species identification using 26S and 16S PCR. After completion of this procedure, the selection of microorganisms from the whole set of species to create a new and less-diverse “$” consortium could take place. The species used were *Brettanomyces bruxellensis*, *Hanseniaspora valbyensis*, *Saccharomyces cerevisiae* and *Pichia occidentalis* for yeasts, and *Acetabocter indonesiensis* and *Oenococcus oeni* for bacteria. This selection was based on the highest population levels and the characterization of species described in the beverage by Tran et al. (2020) [30]. Briefly, cultivated kombucha micro-organisms on different agar media allowed the growth of colonies that were sampled and used for 26S and 16S rRNA PCR identification for yeasts and bacteria, respectively. Following characterization of morphotypes and microscopic observations, unambiguous associations between identities and morphotypes were established and used to discriminate species on agar plates [24].

As carried out in our previous studies [36,39], each population was inoculated at 5 log/mL each on the very first inoculation of a 7-day preliminary fermentation cycle, before cycle 1. Both cultures were transferred as a 125 mL volume unit into Schott^®^ flasks (100 mL working volume, 10 cm bottleneck diameter) from Schott Glaswerke AG (Mainz, Germany) and into Boston bottles (125 mL, 1 cm bottleneck diameter) from Wheaton^®^ (Milville, NJ, USA). The vessel geometry enabled the use of two specific interfacial surfaces, SIS2 “2” and SIS1 “1”, measuring 0.162 cm^−1^ and 0.01 cm^−1^, respectively (Table 1). The two SIS levels were selected by using laboratory glassware with SIS close to those of a cylindrical 1000 L tank for SIS1 (industrial production) and those of a traditional 5 L cylindrical fermentation jar for SIS2 (made-at-home production). In addition, cultures were inoculated with two different methods. In accordance with the Biomère process, a volume of liquid corresponding to 12% of the final volume of a previous culture was used for inoculation. No biofilm was used for inoculation. A second inoculation method, “*”, was studied by inoculating a volume of liquid culture, enabling a cycle to be started with a total acidity equivalent to 10 meq/L. This second inoculation method was only carried out on cultures with SIS1. This resulted in a total of 6 conditions: O1, O*, O2, $1, $* and $2. For each of the different modalities listed, three technical replicates were carried out under the same conditions, and each replicate constituted a backslopping (or propagation) lineage without any mixing taking place at any stage of the study, making a total of 18 lineages.

### 2.2. Condition-Based Culture Backslopping Lineages

The experimental design took the form of 8 consecutive fermentation cycles based on the backslopping principle (inoculation of batch n with batch n-1). A fermentation cycle lasted seven consecutive days, from day 1, the first day of the cycle, to day 7, the last day of the cycle (Figure 1). For all cycles, incubation conditions were static at 26 °C and aerobic, with the bottleneck only protected by gauze pads. The experimental procedure began with a preliminary cycle to prepare the two cultures (original and synthetic), which were subjected to SIS1 and SIS2 on different sets of samples to inoculate the corresponding conditions at the beginning of cycle 1. For example, the original culture incubated with SIS1 = 0.01 cm^−1^ was used to inoculate O1 and O*, and the original culture incubated with SIS2= 0.162 cm^−1^ was used to inoculate O2. The same was applied to the synthetic culture. The preliminary cycle was followed by cycles 1 to 7, during which each lineage of the 6 conditions (O1, O*, O2, $1, $*, $2) was propagated and subjected always to the same parameters (SIS and inoculation mode). Finally, the study ended with cycle 8, where all lineages were subjected to the same inoculation and SIS conditions, i.e., an initial acidity of 10 meq/L and SIS1 = 0.01 cm^−1^. Cycle 8 was used to measure the effect of the biotic factor alone, i.e., any modulation resulting from propagation cycles (1 to 7), as an intrinsic change in microbial dynamics independent of the abiotic factors applied (for example, the SIS or the inoculation method).

### 2.3. Analyses

Various microbiological and physico-chemical parameters were measured on day 7 of each cycle (Figure 1). Biofilm weight was measured using gravimetry directly after the transfer of the whole biofilm (whose diameter ranged between 1 and 10 cm, depending on the vessel) from the liquid surface to a 10 mL sterile 9 g/L NaCl dilution solution using plies. The biofilms were then prepared for microbiological analysis. Cells were extracted by thorough laceration of the biofilm using a sterile scalpel, and the suspension was transferred into a 50 mL tube and vortexed at maximum intensity for 30 s. The resulting suspension was used for plate counting.

Yeast and bacterial species present in the liquid phase and in the solid phase (biofilm) extract were counted on selective solid agar media in order to monitor the evolution of the different populations and species present on the seventh day. On selective WL agar plates from Thermo Fisher Scientific (Waltham, MA, USA), yeast species were differentiated on the basis of a correspondence between the identity and morphotype of the different colonies [24]. This method has been successfully applied to study beer fermentations [41]. Bacteria were cultivated on selective pH 6.2 De Man Rogosa and Sharpe (MRS) agar plates from Condalab (Madrid, Spain) with differentiation of *Oenococcus oeni* from the other species (non-*O. oeni* bacteria) based on characteristic colony size [42]. Incubation occurred at 28 °C in an aerobic condition. The choice of solid agar media and incubation parameters were validated for optimal growth of yeasts and bacteria based on comparative data obtained in a previous study [24]. The choice of a culture-dependent method was based on its higher implementation potential in an industrial context. This allows further fermentation monitoring or in-house R&D based on this methodology using multi-species original or synthetic cultures.

After microbiological analysis, liquid sample underwent centrifugation (3000× *g* for 10 min at 4 °C) to obtain cell-free supernatants. The pH value was measured in the liquid using a Five Easy device equipped with a LE498 sensor, both from Mettler Toledo (Greifensee, Switzerland). Total acidity of kombucha cultures, expressed as meq/L, was measured by titration on days 0, 1, 2, 3, 6 and 7 of the various fermentation cycles using 1N NaOH solution with the addition of a few drops of 0.2% (*v*/*v*) phenolphthalein solution in the sample. The method was adapted from OIV (2009) [43].

Metabolite concentrations were analyzed using high-performance liquid chromatography (HPLC) 260 Infinity II Agilent HPLC (Santa Clara, CA, USA), equipped with a 1260 Quat Pump G7111B and a 1260 G7129A injection module with oven. Acetic and gluconic acid detection occurred at 214 nm thanks to a 1260 DAD WR G7115A module. Sugars (sucrose, glucose, fructose) and ethanol were detected by refractometry using a 1620 RID G7162A module. The separation of the aforementioned metabolites from the 20 µL injection samples was performed with an Aminex^®^ HPX-87H 300 mm/7.8 mm column from Biorad (Hercules, CA, USA) at 30 °C for 35 min in isocratic mode at 0.45 mL/min flow. Mobile phase was 0.065 mmol/L sulfuric acid (H_2_SO_4_). Chromatograms were treated using the OpenLab software (3.4.0 version) from Agilent (Santa Clara, CA, USA).

An acidification coefficient was calculated to reflect the rate of acidification kinetics for each cycle and condition. To do this, a linear regression line was calculated for each replicate from the total acidity values collected on days 2, 3, 6 and 7 over the linearity zone (between days 0 and 2, there is a lag phase [40]). We obtained an equation of the type *y* = *ax* + *b* (where *y* is total acidity in meq/L and *x* is time in days) and a coefficient of determination R^2^ reflecting the quality of the modeling, with a minimum critical threshold set at 0.90. The acidification coefficient is assimilated to the directing coefficient *a*, the unit of which is expressed in meq/L/day.

### 2.4. Data Treatment

Linear regression and standard deviation calculations were performed using Excel 2021 software from Microsoft (2310 version). Non-parametric Kruskal–Wallis tests were performed to assess the presence of a significant difference between the values (biological replicates, *n* = 3, α = 5%). If the null hypotheses were rejected, a Dunn post hoc test with Benjamini–Hochberg correction was performed [44]. Principle Component Analysis (PCA) was also performed in association with ascending hierarchical classification analysis. All analyses were performed using R software (4.2.2 version) [45] and the following packages: factoextra (1.0.7) [46], FactoMineR (2.7) [47] and FSA (0.9.4) [48].

## 3. Results and Discussion

Preliminarily, the microbial composition of the original “O” culture based on isolation and identification of grown colonies was established as described in Table 1. E-values associated with the reliability of species identification were all lower than 0.001 and were associated with pairwise identities superior to 89.3% of sequences ranging between 495 and 607 base pairs, with query coverage all equal to 100%. Thus, compared to the “$” synthetic culture, the “O” culture possessed at least one additional yeast species (*Zygotorulaspora florentina*) and two additional lactic acid bacteria species (*Liquorilactobacillus mali* and *Liquorilactobacillus satsumensis*), based on a culture-dependent method.

The first step was to assess in cycle 8 whether the consortia had indeed undergone modulation after the previous seven cycles (Table 2 and Table 3). Parameters showing significant differences were sucrose, fructose, total sugars, ethanol concentrations, *S. cerevisiae* and *H. valbyensis* populations in the liquid phase and the *P. occidentalis* population in the biofilm. Two significant differences point to the effect of culture type or production parameters. Firstly, the ethanol content was significantly higher in the $2 condition compared with O2. This shows that application of a higher SIS on cycles 1 to 7 induced lower ethanol accumulation for the original culture (0.3 g/L versus 0.5 g/L), whereas both cultures were subjected to the same lower SIS. Secondly, there was a significant difference between conditions O2 and $1. Condition O2 had significantly higher concentrations of total sugars (32.7 g/L versus 22.4 g/L), sucrose (17.0 g/L versus 0.8 g/L), a significantly lower ethanol concentration (2.5 g/L versus 5.0 g/L) and a significantly lower *S. cerevisiae* population than $1 (5.4 log(CFU)/mL versus 5.9 log(CFU)/mL). These results link the *S. cerevisiae* population level to a higher intensity of sucrose hydrolysis and ethanol accumulation. Although the analysis of cycle 8 allows for the assessment of modulation, it offers only a partial window into the biological phenomena that may have taken place during the different cycles.

In order to visualize the evolution of the numerous analytical parameters for all conditions, a Principal Component Analysis (PCA) was performed on all the data collected (Figure 2). On the sample plot, the distribution of samples is based on dimensions 1 and 2, with an eigenvalue of 63.64%, indicating a high level of data explanation (Figure 2A). The ellipses, corresponding to the ascending hierarchical classification, separate the conditions into three clusters: a cluster including the six conditions at cycle 8; a cluster including all conditions from the original “O” culture except O1 and O* at cycle 7, and also including conditions from the synthetic cultures $1, $* and $2 at cycle 1 only; and finally a cluster containing all the other conditions, including all the conditions resulting from synthetic cultures between cycles 2 and 7.

These observations show that at cycle 1, the fermentative behavior of the six conditions was comparable, but subsequently diverged. Indeed, from cycle 2 onwards, conditions including the synthetic “$” culture changed their behavior (switching from one cluster to another), leading to a lasting divergence from cultures including the original “O” culture up to and including cycle 6. In cycle 7, conditions O1 and O* adopted a similar behavior to conditions including the synthetic culture, and in cycle 8, all conditions converged to form a new cluster, indicating a new fermentative behavior. Importantly, cycle 8 stood out most strongly from the other cycles.

The content of the changes in fermentative behavior is explained by the plot vector, which projects the analytical parameters as vectors, translating their correlation according to dimensions 1 and 2 (Figure 2B). Analysis of the contribution of analytical parameters shows that, for a score of over 70%, dimension 1 is described positively by total sugars, sucrose and biofilm microbial populations (excluding *S. cerevisiae*) and negatively by the content of monosaccharides (glucose and fructose), ethanol and the *S. cerevisiae* population in the liquid phase. Dimension 1 therefore reflects sucrose hydrolysis and alcoholic fermentation, establishing a clear correlation with the *S. cerevisiae* population. No parameter has a score of over 70% to describe dimension 2. It should be noted, however, that the study of correlations between parameters makes it possible to associate the production of monosaccharides from sucrose hydrolysis with the acidification caused by the production of acetic and gluconic acids. We thus find the acidification process known for kombucha [30]. It can be noted that the vectors corresponding to the microbial populations in the biofilm are all highly correlated. This can be explained by the extraction stage of the biofilm cells, which had a direct impact on the counts: the more efficient the extraction, the higher the population counts. So, the analysis highlights this bias. The fact that populations in the biofilm decreased over the cycles was due to the decrease in biofilm weight. This can be explained by visual observations: the increase in carbon dioxide emissions in the liquid phase hindered the optimal establishment of the biofilm (notably the presence of gas pockets).

Thus, the change in behavior of the conditions studied can be described as a transition from low yeast activity to higher activity via intermediate activity, characterized by the intensification of sucrose hydrolysis and alcoholic fermentation. This intensification was correlated with an increase in the *S. cerevisiae* population in the liquid phase only. However, this intensification of yeast activity was not accompanied by an intensification of acidification.

Interpreting variations of the key parameters highlighted above enables us to investigate the existence of causal relationships. In Figure 3, we observe an overall decrease in total sugar and sucrose content over the cycles and an increase in monosaccharide content as a result of sucrose hydrolysis. The divergence in behavior between cycles 2 and 6 between conditions derived from different cultures was clearly visible, particularly regarding sucrose and fructose contents (Figure 3A,D). There was also a general intensification of hydrolysis between cycles 7 and 8. Figure 4 shows changes in the ethanol production and acidification kinetics. There was little variation in ethanol concentration between cycles 1 and 7. However, $1 ethanol levels were significantly higher than those of $2 in cycles 3 (2.3 g/L versus 1.1 g/L), 4 (3.2 g/L versus 2.2 g/L) and 8 (5.0 g/L versus 4.2 g/L), suggesting the role of SIS in providing the oxygen required for ethanol oxidation by acetic acid bacteria (Figure 4A). In addition, ethanol content increased between cycles 7 and 8, except for the O2 condition. Acidification kinetics showed little variation for conditions derived from the original “O” culture and for $*, probably due to the inoculation process, whereas they were chaotic for conditions $1 and $2 derived from the synthetic “$” culture (Figure 4B).

Figure 5 shows the population variations of *S. cerevisiae* and *H. valbyensis* in the liquid phase, which exhibited significant differences at cycle 8. It can be seen that the *S. cerevisiae* population was below or equal to the detection limit of 4 log(CFU)/mL for conditions from the original “O” culture between cycles 1 and 3 for O1 and O2, and between cycles 1 and 5 for O*. The $1 population of *S. cerevisiae* was significantly higher than O* from cycle 2 to cycle 4 (Figure 5A). From cycle 7 onwards, the *S. cerevisiae* population generally increased for all conditions. For *H. valbyensis*, a population drop of up to 2 log was observed between cycles 2 and 5 for conditions $1 and $*, compared with cycle 1 (Figure 5B). It reached the detection threshold in cycle 5 for $1 and cycle 6 for $*. Populations for the other conditions remained between 5.5 and 7 log/mL between cycles 1 and 7. This observation suggests that the maintenance of the *H. valbyensis* population was dependent on access to oxygen for conditions derived from the synthetic “$” culture only. This observation is supported by the decrease in population between cycles 7 and 8 for condition $2 due to the change from SIS2 to SIS1 (higher and lower access to oxygen, respectively).

Extending the observation to all the parameters described, we can highlight the existence of several concomitant significant differences (Table A1). The increase in the *S. cerevisiae* population was associated with a decrease in sucrose content and an increase in fructose content (between O* and $1 for cycles 2 and 3). A set of concomitant significant differences can also be identified, associating lower sucrose and higher ethanol concentrations with a lower *H. valbyensis* population (O2 versus $1 for cycles 5 and 7).

To sum up, the application of the different conditions over seven cycles did not result in modulation of microbial populations and activity according to these conditions at cycle 8. Yet, behaviors evolved along the cycles, although they were similar in cycle 1 and converged in cycles 7 and 8. As opposed to abiotic parameters, the impact of the culture type (biotic parameter) was visible in the speed of the microbial modulation over the cycles. It was faster for conditions resulting from the synthetic “$” culture, but the rate of acidification varied unpredictably (Figure 4). A recent study reported similar opposition between the original consortium and the synthetic consortium associated, respectively, with a more stable and more unpredictable endpoint pH value per cycle in Mabisi fermentation [23]. According to the literature, synthetic microbial communities appear to be associated with chaotic behavior, i.e., an extreme sensitivity to initial environmental parameters [49,50]. In contrast, a microbial community such as the original “O” culture with higher biodiversity is associated with greater stability of microbial communities and their functionalities [51,52,53,54]. On a lower scale, it was also observed that less ethanol was produced in the O2 condition than in the $2 condition at cycle 8, as a result of modulation depending on the culture type. Regarding the inoculation procedure, it is important to note that the inoculation ratio in terms of microbial populations cannot be controlled and could be the source of batch-to-batch disparities. However, a recent study compared sweet tea cultures with different inoculation ratios between yeasts and acetic acid bacteria with five species of each kingdom [14]. The results showed that population equilibrium was achieved at all inoculation ratios tested. Microbial activities were affected, mainly yeast activities (sucrose hydrolysis and ethanol production). Acetic acid production kinetics, on the other hand, were equivalent. This phenomenon was present, though less marked, when a pairing of one species of yeast and one species of acetic bacteria was used. According to Huang et al. (2022), it seems, therefore, that the inoculation rate between yeasts and bacteria is not a determining factor if the cultures have a certain level of microbial diversity [14].

In the present study, production parameters had no marked effect in terms of modulation. However, this was the case for other matrices and with different parameters, such as temperature or pH [18,19,20]. Standardization of initial total acidity showed no effect compared to inoculation with a fixed volume percentage of 12% (*v*/*v*). The observed limitation of the SIS effect can be explained by the fact that sucrose hydrolysis can act as the limiting reaction in the fermentation process. In other words, it is the rate of production of monosaccharides, the precursors of organic acids, that dictates the rate of acidification, not the SIS. Conditions including the original culture illustrated this well, particularly in cycles 1 to 3, when monosaccharide levels were at the detection limit (0.1 g/L; Figure 3). This reflected a strong tension on the use of organic acid precursors (acetic and gluconic acids), making the role of SIS in acidification kinetics imperceptible under these conditions. The effect of SIS would be perceptible if the precursors were to accumulate further, as was the case for conditions resulting from the synthetic “$” culture from cycle 2 onwards and under the activity of superior *S. cerevisiae* populations (Figure 3 and Figure 4A). From another perspective, acetic acid bacteria have to deal with the limitation of available oxygen or carbon substrates, either of which is limiting depending on SIS conditions and other microbial activities.

The effects of SIS mainly affected ethanol content in conditions derived from the synthetic “$” culture, with slightly lower concentrations in conditions subjected to a higher SIS and access to oxygen. Lower ethanol concentrations were associated with higher *H. valbyensis* populations and vice versa. Indeed, ethanol accumulation appeared detrimental to *H. valbyensis*, causing its population to decline. Moreover, a decrease in the *H. valbyensis* population during phase 2 of kombucha production has been reported in our previous studies [30,39]. According to [55], this species is sensitive to ethanol based on cultivation in media enriched with ethanol. The production of ethanol was due to the increase in *S. cerevisiae* population and fermentative activity. The consequences for the product were lower sugar levels and ethanol accumulation without any significant acceleration of acidification. In addition to the regulatory problems associated with ethanol content, this has heavy consequences for the product’s sugar/acid balance, a key sensory parameter [29]. However, it should be noted that the maximum ethanol levels observed (close to 5 g/L) remained below the regulatory threshold of 12 g/L in the European Union legislation [56]. This phenomenon corroborates the observations made in an industrial production context, and a similar phenomenon has been reported during the monitoring of sourdough backslopping cycles [57]. It is noteworthy that several one-batch investigations of kombucha fermentation report the *S. cerevisiae* population as minor, in contrast to its typical dominance in alcoholic beverage fermentations [24,30,37,39,58]. Therefore, this species appears to be less adapted to sugared tea compared to *Brettanomyces bruxellensis*, for example [24,30,37,39,59].

With the current data, it is difficult to explain why the *S. cerevisiae* population increased with each cycle. One hypothesis would be that the frequency of cycles induced the establishment of environmental conditions that benefited the progressive implantation of this species, in line with its fitness. This backslopping process would be opposed, for example, to the use of a mother culture that is much more acidic and less favorable to the growth of *S. cerevisiae*. We could also mention the accumulation of beneficial metabolites or nutrients, such as assimilable nitrogen content, or a selection bias due to the inoculation method. To date, there is little scientific evidence linking invertase activity to fermentative activity beyond catabolic repression by glucose [31]. Indeed, expression of the *SUC2* gene encoding an intracellular invertase and a periplasmic invertase is dependent on the SNF1 protein complex regulating energy homeostasis at the cellular level [60]. In addition, the URE2/GLN3 system has been described as regulating invertase and asparaginase activity in *S. cerevisiae* depending on available nitrogen [61]. Further investigations are needed to assess those hypotheses, including assimilable nitrogen analysis or even metabolomics.

## 4. Conclusions

This study highlighted the existence of phases in microbial activity as propagation cycles progressed. The transitions between phases occurred faster for the synthetic consortium, composed of a few species isolated from the original kombucha consortium. This led to fermentation kinetics that were reproducible over several cycles but that could also deviate and shift abruptly to different behaviors. These changes were mainly induced by an increase in the *S. cerevisiae* population, associated with an intensification of sucrose hydrolysis, sugar consumption and an increase in ethanol content, without any significant acceleration in the rate of acidification. This corroborates on-site observations and calls for a revision of the inoculation methods, using mother cultures, for example. These observations also underline the difficulty of controlling reproducibility, particularly when production parameters vary, given that the response of consortia in terms of fermentation kinetics is not necessarily immediate.

On an applicative level, this study suggests that backslopping as an inoculation method induces undesirable changes in behavior in kombucha cultures, as observed in the production context. It points to subtle changes in the balance of yeast populations, whose activity is key to fermentation kinetics. In addition, it also highlights a link between the speed of response to changes in production conditions, and hence reproducibility, and the biodiversity of the kombucha culture. A decrease in the species number of a kombucha culture was highlighted over several years in a previous study, following the absence of thermal regulation on the production site [24]. Taken together, this suggests that the reproducibility of kombucha fermentations relies on high biodiversity to slow down the modulations of microbial dynamics induced by the sustained rhythm of backslopping cycles. Otherwise, producers are exposed to drifts in microbial activity that can lead to higher ethanol production in phase 1, thus increasing the risk of exceeding the regulatory threshold during the second fermentation phase.

As a promising study model for microbial ecology, kombucha consortia could be implemented in further research. Similar experimental procedures based on backslopping including metagenomics, could be used to gain further insight into the microbial dynamics. In addition, those experiments could test other production parameters such as fermentation temperature, water quality or initial assimilable nitrogen and simulate adverse events such as exogenous microbial invasion.

## Figures and Tables

**Figure 1 foods-13-01181-f001:**
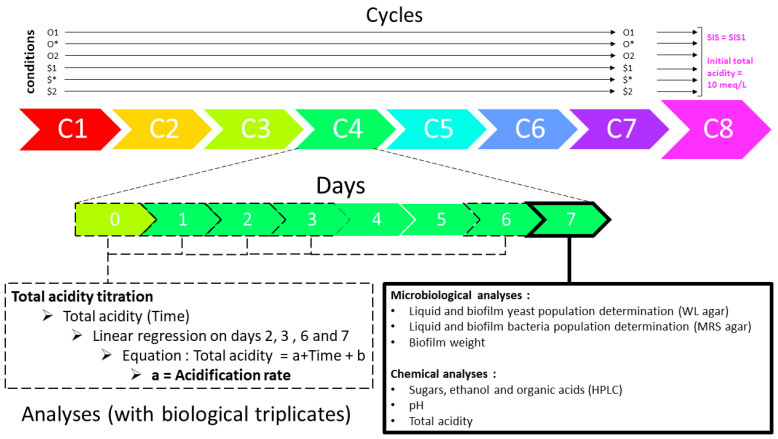
Diagram detailing the experimental design and methodology of the study. Inoculation was performed with original culture (O) or synthetic culture ($) culture with 12% (*v*/*v*) liquid culture or an alternative inoculation method used to set the initial total acidity of 10 meq/L for each cycle (*). Specific Interfacial Surface (SIS) 1 or 2 was applied. Eight backslopped cycles (C) have been performed from C1 to C8. Each cycle lasted seven consecutive days. The day-to-day analysis scheme is detailed for C4 only and was applied for each cycle.

**Figure 2 foods-13-01181-f002:**
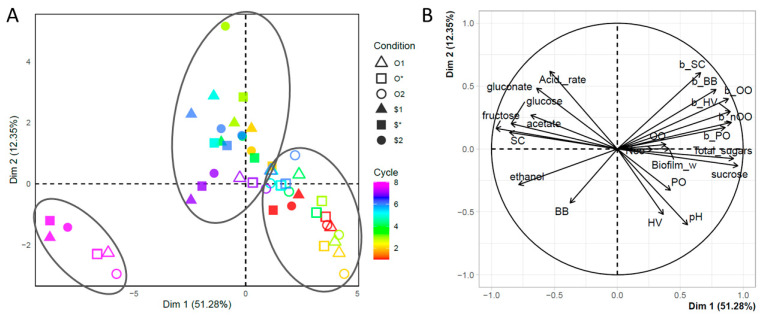
Visualization of data by Principal Component Analysis according to (**A**) a sample plot and (**B**) a vector plot. The ellipses correspond to the results of the ascending hierarchical classification analysis. BB = *Brettanomyces bruxellensis*, HV = *Hanseniaspora valbyensis*, SC = *Saccharomyces cerevisiae*, PO = *Pichia occidentalis*, *OO = Oenococcus oeni*, *nOO = non-Oenococcus oeni*. If the sample code is preceded by “b_”, it refers to the microbial count in the biofilm, otherwise to the liquid phase.

**Figure 3 foods-13-01181-f003:**
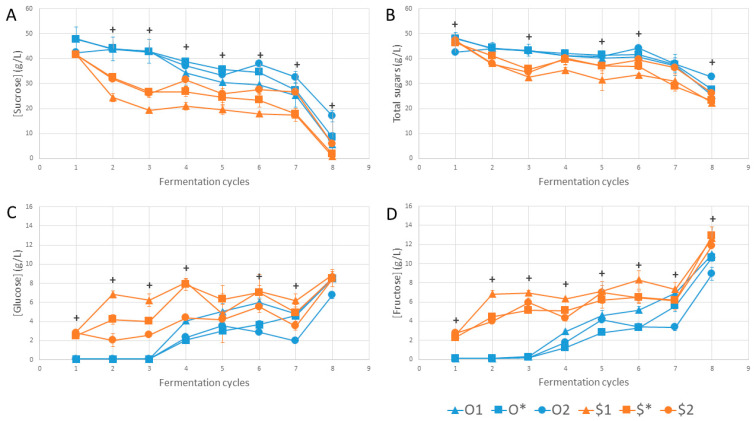
Variation in sugar content: (**A**) sucrose, (**B**) total sugars, (**C**) glucose and (**D**) fructose over cycles. “+” symbols correspond to the existence of at least one significant difference between values within conditions for a given cycle according to the Kruskal–Wallis test (*n* = 3, α = 0.05). Error bars correspond to standard deviation.

**Figure 4 foods-13-01181-f004:**
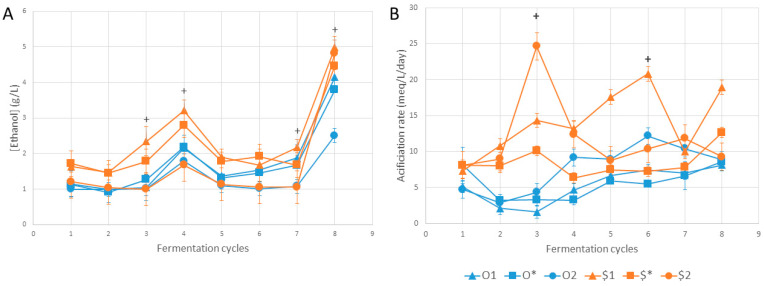
Variations in (**A**) ethanol content and (**B**) acidification over cycles. “+” symbols correspond to the existence of at least one significant difference between values within conditions for a given cycle according to the Kruskal–Wallis test (*n* = 3, α = 0.05). Error bars correspond to standard deviation.

**Figure 5 foods-13-01181-f005:**
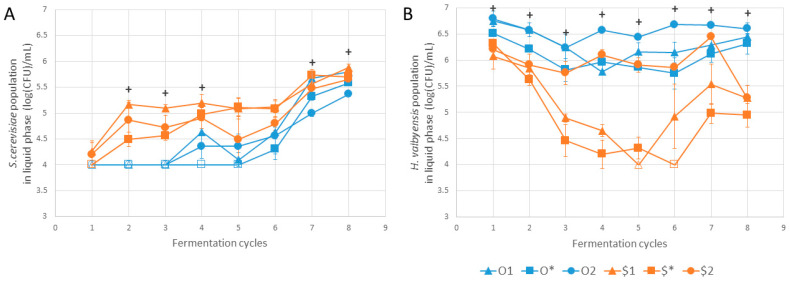
Variation in populations of (**A**) *Saccharomyces cerevisiae* and (**B**) *Hanseniaspora valbyensis* over cycles. Populations under the detection limit of 4 log(CFU)/mL are indicated by unfilled symbols. “+” symbols correspond to the existence of at least one significant difference between values within conditions for a given cycle according to the Kruskal–Wallis test (*n* = 3, α = 0.05). Error bars correspond to standard deviation.

**Table 1 foods-13-01181-t001:** Summary of conditions tested during the study according to factors: type of container; SIS; inoculation method.

Culture	Microbial Composition	Bottle Type	SIS	Inoculation	Code
Original	*Brettanomyces bruxellensis*, *Hanseniaspora valbyensis*, *Saccharomyces cerevisiae*, *Pichia occidentalis*, *Zygotorulaspora florentina*, *Oenococcus oeni*, Monitored as non-*O. oeni*: *Acetobacter indonesiensis*, *Liquorilactobacillus mali*, *Liquorilactobacillus satsumensis*	Boston	SIS1	12% (*v*/*v*)	O1
		Boston	SIS1	*: volume for fixed initial total acidity of 10 meq/L	O*
		Schott^®^	SIS2	12% (*v*/*v*)	O2
Synthetic	*Brettanomyces bruxellensis*, *Hanseniaspora valbyensis*, *Saccharomyces cerevisiae*, *Pichia occidentalis*, *Oenococcus oeni*, Monitored as non-*O. oeni*: *Acetobacter indonesiensis*	Boston	SIS1	12% (*v*/*v*)	$1
		Boston	SIS1	*: volume for fixed initial total acidity of 10 meq/L	$*
		Schott^®^	SIS2	12% (*v*/*v*)	$2

All samples were produced as biological triplicates. SIS1 = 0.01 cm^−1^ and SIS2 = 0.162 cm^−1^.

**Table 2 foods-13-01181-t002:** Average values of the parameters analyzed in the liquid phase for each of the 6 conditions studied in Cycle 8, together with the *p*-value returned by the Kruskal–Wallis test (*n* = 3, α = 0.05). Common letters indicate absence of significant difference according to Dunn’s test with Benjamini-Hochberg adjustment.

Condition	Sucrose (g/L)	Glucose (g/L)	Fructose (g/L)	Total Sugars (g/L)	Ethanol (g/L)	Gluconate (g/L)	Acetate (g/L)	Acidification Rate (meq/L/day)	*B. bruxellensis* (log(CFU)/mL)	*H. valbyensis* (log(CFU)/mL)	*S. cerevisiae* (log(CFU)/mL)	*P. occidentalis* (log(CFU)/mL)	*O. oeni* (log(CFU)/mL)	Non-*O. oeni* (log(CFU)/mL)
O1	5.7 ab	8.6	11.0 ab	25.3 ab	4.2 ab	1.4	0.2	8.1	6.8	6.4 a	5.8 a	4.0	5.1	5.7
O*	8.4 ab	8.5	10.6 ab	27.5 ab	3.8 ab	3.3	0.2	8.6	6.7	6.3 a	5.6 ab	4.0	5.6	5.6
O2	17.0 b	6.8	8.9 b	32.7 b	2.5 b	2.4	0.2	8.9	6.8	6.6 a	5.4 b	4.0	5.3	5.7
$1	0.8 a	8.9	12.7 a	22.4 a	5.0 a	2.4	0.3	18.9	6.7	5.3 a	5.9 a	4.0	5.2	5.4
$*	1.7 a	8.5	12.9 a	23.2 a	4.5 ab	4.4	0.3	12.7	6.7	4.9 a	5.7 ab	4.0	5.1	5.5
$2	5.8 ab	8.4	11.9 ab	26.1 ab	4.8 a	3.8	0.3	9.3	6.6	5.3 a	5.7 b	4.0	5.4	5.5
*p*-value	0.01002	0.1904	0.01072	0.01471	0.008749	0.17	0.514	0.1465	0.4569	0.0149	0.01779	Not calculable	0.2489	0.4012

**Table 3 foods-13-01181-t003:** Average values of the parameters analyzed in the biofilm for each of the 6 conditions studied in Cycle 8, together with the *p*-value returned by the Kruskal–Wallis test (*n* = 3, α = 0.05). Common letters indicate absence of significant difference according to Dunn’s test with Benjamini-Hochberg adjustment.

Condition	Biofilm Weight (mg)	*B. bruxellensis* (log(CFU)/g Fresh Biofilm)	*H. valbyensis* (log(CFU)/g Fresh Biofilm)	*S. cerevisiae* (log(CFU)/g Fresh Biofilm)	*P. occidentalis* (log(CFU)/g Fresh Biofilm)	*O. oeni* (log(CFU)/g Fresh Biofilm)	Non-*O. oeni* Bacteria (log(CFU)/g Fresh Biofilm)
O1	1.9	7.2	5.7	6.0	2.5 a	7.0	8.1
O*	31.4	6.4	5.2	5.0	3.5 a	5.8	7.3
O2	41.2	6.3	5.2	4.7	3.1 a	5.4	6.9
$1	94.0	5.3	4.3	4.7	2.8 a	5.3	7.7
$*	29.3	6.0	4.6	4.8	2.7 a	5.5	7.1
$2	23.0	6.3	5.2	5.5	2.6 a	5.6	7.0
*p*-value	0.1109	0.101	0.06201	0.06567	0.03653	0.2255	0.5052

## Data Availability

The original contributions presented in the study are included in the article, further inquiries can be directed to the corresponding author.

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
