# Peer review of "Identification of Key Parameters Inducing Microbial Modulation during Backslopped Kombucha Fermentation"

_foods, 2024, doi:10.3390/foods13081181_

Round 1

Reviewer 1 Report

Comments and Suggestions for Authors

This work evaluates the impact of a number of abiotic and biotic factors on the production of kombucha. This is a key challenge in the fermented foods field, and is a well-executed and well-described piece of work that will be of interest to those working in this area, and also commercial producers. I think the manuscript should be accepted for publication, but I have some small comments below that I think will improve the experience and pleasure of reading this work.

General comments:

Table 1: Do we know if there are multiple strains of these species in the community, or only one of each? The species identification was only based on 26S or 16S? I think a sentence should be added to confirm this. Is it possible that metagenetics or metagenomics analyses would detect other microbes in the original culture? There were only a limited selection of media and growth conditions used for selecting yeasts and bacteria (in particular). This is a key limiting factor in the publication. The estimation of the microbes in the fermentation is based on colony morphology, and is subject to error. An image of the plates showing variation in morphology might help to convince the reviewer this is valid. Multiple strain variation in a community would help with robustness in response to stressors (as opposed to a synthetic community)>

There is no explanation provided as to why the authors omitted the microbes they did in the synthetic community.

Comments on the Quality of English Language

Unfortunately, I do think some assistance in terms of the written English would help improve the manuscript. This will not be exhaustive, a native English speaking colleague could help, but some of the words and the sentence construction could do with some work.

For example:

“Led by fermented beverages such 33 as kefir and kombucha; changes in consumption behavior now occur worldwide and calls 34 for modifications in regulatory frameworks” doesn’t make sense.

Editing of “This movement stemmed in the last decades from western countries and propagated worldwide thanks to globalized markets 36 [5,6], trending values supporting well-being and self-appropriation of nutrition [7] and 37 increasing amount of scientific evidence highlighting effective benefits of fermented food 38 consumption along with the unravelling of their mechanisms” would improve this significantly.

“All treatments were performed using R software (4.2.2 version).” Should be “All analyses were performed using R software (4.2.2 version).”

L225:“the analysis of cycle 8 allows to state on the existence of modulations” I don’t understand what this means.

Author Response

R1-01 : This work evaluates the impact of a number of abiotic and biotic factors on the production of kombucha. This is a key challenge in the fermented foods field, and is a well-executed and well-described piece of work that will be of interest to those working in this area, and also commercial producers. I think the manuscript should be accepted for publication, but I have some small comments below that I think will improve the experience and pleasure of reading this work.

A-01 : We thank the reviewer for the time spent reviewing our manuscript and for the feedbacks. We will answer the questions point-by-point (A1-01, A1-02 etc..).

General comments:

R1-02 : Table 1: Do we know if there are multiple strains of these species in the community, or only one of each? The species identification was only based on 26S or 16S? I think a sentence should be added to confirm this. Is it possible that metagenetics or metagenomics analyses would detect other microbes in the original culture? There were only a limited selection of media and growth conditions used for selecting yeasts and bacteria (in particular). This is a key limiting factor in the publication. The estimation of the microbes in the fermentation is based on colony morphology, and is subject to error. An image of the plates showing variation in morphology might help to convince the reviewer this is valid. Multiple strain variation in a community would help with robustness in response to stressors (as opposed to a synthetic community)>

A1-02 : Regarding the detection of strains : The differentiation of strains is indeed another layer of investigation. Based on our previous experiments in the lab, this necessitates additional methodology development for non-standard species (for example acetic acid bacteria). Given the initial complexity in terms of species across 2 different kingdoms, the diversity in strain was not investigated in the present study. Moreover, estimation of strain populations was not feasible given the amount of plates and colonies involved in the experimental design.

The identification of species was performed using 26S and 16S only, with primers chosen for the most efficient species discrimination according to our previous study Mas et al. (2022)*. A statement was added in-text L106-L111.

“Briefly, cultivated kombucha micro-organisms on different agar media allowed the growth of colonies that were sampled and used for 26S and 16S identification for yeasts and bacteria respectively. Following characterization of morphotypes and microscopic observations, unambiguous associations between identities and morphotypes were established and used to discriminate species on agar plates [24].”

Metagenomic analysis could have been approach for this investigation and allowed to identify non-culturable microbes. However, culture-dependent method was chosen to allow the implementation of the methodology in the brewery. This opens perspectives for multi-species monitoring of fermentations whether using original of synthetic cultures. A statement was added in-text regarding this matter (L178-L181) :

“The choice of a culture-dependent method was based on higher implementation potential in industrial context. This allows further fermentation monitoring or in-house R&D based on this methodology using multi-species original or synthetic cultures.”

The detail of the morphology-identity association method is described in Mas et al. (2022) and includes images. This study is cited multiple times in the main text and namely L178. We believe it would be quite redundant to display them again.

R1-03 : There is no explanation provided as to why the authors omitted the microbes they did in the synthetic community.

A1-03: Modification was made in-text L105-106 to justify this selection : “This selection was based on highest population levels and characterization of species described in the beverage by Tran et al. (2020).”

Comments on the Quality of English Language

R1-04 : Unfortunately, I do think some assistance in terms of the written English would help improve the manuscript. This will not be exhaustive, a native English speaking colleague could help, but some of the words and the sentence construction could do with some work.

For example:

“Led by fermented beverages such 33 as kefir and kombucha; changes in consumption behavior now occur worldwide and calls 34 for modifications in regulatory frameworks” doesn’t make sense.

Editing of “This movement stemmed in the last decades from western countries and propagated worldwide thanks to globalized markets 36 [5,6], trending values supporting well-being and self-appropriation of nutrition [7] and 37 increasing amount of scientific evidence highlighting effective benefits of fermented food 38 consumption along with the unravelling of their mechanisms” would improve this significantly.

“All treatments were performed using R software (4.2.2 version).” Should be “All analyses were performed using R software (4.2.2 version).”

L225:“the analysis of cycle 8 allows to state on the existence of modulations” I don’t understand what this means.

A1-04: Modifications of the phrasing were made L33-39 and L238-L241.

“Fermented beverages such as kefir and kombucha led to changes in consumption behavior that now occur worldwide, which calls for modifications in regulatory frameworks [4]. This movement stemmed in the last decades from western countries and propagated worldwide thanks to globalized markets [5,6], trending values supporting well-being and self-appropriation of nutrition [7]. Moreover, increasing amount of scientific evidence highlighted effective benefits of fermented food consumption along with the unraveling of their mechanisms [8–11]"

“Although the analysis of cycle 8 allows to highlight modulations effects, it offers only a partial reading window on the biological phenomena that may have taken place during the different cycles.”

Reviewer 2 Report

Comments and Suggestions for Authors

Manuscript 2905771

The contribution assesses the impact of production parameters on the reproducibility of kombucha fermentation over several production cycles based on backslapping. To this objective, biofilm weight, key physicochemical parameters, fermentation substrates, and metabolites were controlled. The experiment included original and synthetic consortiums, two specific interfacial surfaces (SIS), and two inoculation methods (only for the reduced SIS). The study was detailed, but the effects can hardly establish a relationship between cause and effect because of the chaotic behaviour in most cases. I would suggest introducing some clarification in the title, e.g., “Tentative identification of the causes regarding the microbial modulation…..”

Abstract

The abstract is somewhat confusing and omits some treatments, e.g., the two inoculation methods, which could be necessary to explain the six treatments mentioned. Some clarification could improve understanding of the experimental design and the results.

Methods

Discrimination on agar plates just based on colony morphotype characterization deserves more extensive justification.

How the selected two SIS levels were selected.

Table 1 requires simplification. For example, the column replicates could be omitted and mentioned only in the text or in a footnote. Including the microbial composition as a column also introduces hard-to-read information. Explaining it as a footnote or text could simplify the exposition of the design and focus attention on the variables involved in the design. More details could be included in footnotes or text. Additionally, decimal points would be introduced.

Figure 1 is also not clear. The information on the analysis performed is not easily related to the information intended to be transmitted. It is suggested to comment on text. Also, relate C4 to days may led to confusion regarding the other cycles. Maybe a simple note in the figure caption could suffice. “Each cycle lasted 7 days….”

Data treatment

All treatments, statistical analyses, studies, etc.

The R packages used for the analysis should be included in this section. Besides, citations for them and the R software should be included.

Table 2. The current structure is confusing. It is suggested that the biofilm be split into two tables so that it can appear separately. In this way, the word “Biofilm could be removed from the headings and improve its understanding.

Table 2- Some expressions on the heading are not clear (e.g. which is the meaning of log/mL)

Table 2. Some information (by cell or pulled) on variability 8 may in parenthesis) could be convenient for visually supporting differences observed.

How could it affect the unit of the values to the comparison between treatments, particularly in those expressed as log?

Figures 3, 4 and 5. The use of stars for the significant differences may be confused with the same sign used for identifying the treatments with a fixed initial total acidity of 10 meq/L

Discussion and conclusions

Pay attention in these two sections to the inevitable influences of non-controlled unexpected factors, which are always randomly present in natural products. The comments are excessively detailed cycle by cycle. Focus on trends.

Comments on the Quality of English Language

Some lenguage revision requiered.

Author Response

R2-01: The contribution assesses the impact of production parameters on the reproducibility of kombucha fermentation over several production cycles based on backslapping. To this objective, biofilm weight, key physicochemical parameters, fermentation substrates, and metabolites were controlled. The experiment included original and synthetic consortiums, two specific interfacial surfaces (SIS), and two inoculation methods (only for the reduced SIS). The study was detailed, but the effects can hardly establish a relationship between cause and effect because of the chaotic behaviour in most cases. I would suggest introducing some clarification in the title, e.g., “Tentative identification of the causes regarding the microbial modulation…..”

A2-01: We thank the reviewer for the time spent reviewing our manuscript and for the feedbacks. We will answer the questions point-by-point (A-01, A-02 etc..).

The effects observed were not chaotic, as it is clearly demonstrated based on statistical analysis that the propagation itself is the origin of modulation, as production parameters such as SIS and inoculation rate induced weaker effects.

Title (L2-L3) was changed to : “Identification of key parameters inducing microbial modulation during backslopped kombucha fermentation”

Abstract

R2-02: The abstract is somewhat confusing and omits some treatments, e.g., the two inoculation methods, which could be necessary to explain the six treatments mentioned. Some clarification could improve understanding of the experimental design and the results.

A2-02: The abstract was modified while keeping the word-count below the limit of 200 words (L12-L15):

“Six conditions with varying oxygen accessibility (specific interface surface) and initial acidity (through the inoculation method) of the cultures were carried out and compared to an original kombucha consortium and a synthetic consortium assembled from yeasts and bacteria isolated from the original culture.”

Methods

R2-03: Discrimination on agar plates just based on colony morphotype characterization deserves more extensive justification.

A2-03: The detail of the morphology-identity association method is described in Mas et al. (2022)*. The following text was added L105-L110:

“Briefly, cultivated kombucha micro-organisms on different agar media allowed the growth of colonies that were sampled and used for 26S and 16S rRNA PCR identification for yeasts and bacteria respectively. Following characterization of morphotypes and microscopic observations, unambiguous associations between identities and morphotypes were established and used to discriminate species on agar plates.”

Also, the following text with reference was added L172-L173 :

“This method has been successfully applied to study beer fermentations [41].”

R2-04: How the selected two SIS levels were selected.

A2-04: The following statement was added L118-L121 :

“The two SIS levels were selected by using laboratory glassware with SIS close to those of a cylindrical 1000L tank for SIS1 (industrial production), and those of a traditional 5L cylindrical fermentation jar for SIS2 (made-at-home production)”

R2-05: Table 1 requires simplification. For example, the column replicates could be omitted and mentioned only in the text or in a footnote. Including the microbial composition as a column also introduces hard-to-read information. Explaining it as a footnote or text could simplify the exposition of the design and focus attention on the variables involved in the design. More details could be included in footnotes or text. Additionally, decimal points would be introduced.

A2-05: The suggested modification were made L131-L132. Erasing the “replicate” column allows better readability of the microbial compositions.

R2-06: Figure 1 is also not clear. The information on the analysis performed is not easily related to the information intended to be transmitted. It is suggested to comment on text. Also, relate C4 to days may led to confusion regarding the other cycles. Maybe a simple note in the figure caption could suffice. “Each cycle lasted 7 days….”

A2-06: The figure and its caption were modified L153-L157 to convey information more efficiently.

“Figure 1. Diagram detailing the experimental design and methodology of the study. Inococulation was performed with original culture (O) or synthetic culture ($) culture with 12 % (v/v) liquid culture or an alternative inoculation methods used to set the initial total acidity of 10 meq/L for each cycle (*). Specific Interfacial Surface (SIS) 1 or 2 was applied. Each cycle lasted seven consecutive days.  The day-to-day analysis scheme is detailled for C4 only and was applied for each cycle.”

Data treatment

All treatments, statistical analyses, studies, etc.

R2-07: The R packages used for the analysis should be included in this section. Besides, citations for them and the R software should be included.

A2-07: Packages were added L213-L215 and appropriate citations were added.

“All analyses were performed using R software (4.2.2 version) [44] and the following packages : factoextra (1.0.7) [45], FactoMineR (2.7) [46] and FSA (0.9.4) [47].”

R2-08: Table 2. The current structure is confusing. It is suggested that the biofilm be split into two tables so that it can appear separately. In this way, the word “Biofilm could be removed from the headings and improve its understanding.

Table 2- Some expressions on the heading are not clear (e.g. which is the meaning of log/mL)

Table 2. Some information (by cell or pulled) on variability 8 may in parenthesis) could be convenient for visually supporting differences observed.

How could it affect the unit of the values to the comparison between treatments, particularly in those expressed as log?

A2-08: The tables were split according to the suggestion at pages 8 and 9. The unit was changed to “log(CFU)/mL”. We believe that the addition of standard deviation would impair the readability of the tables. The letters associated to the statistical analysis were put as superscripts.

R2-09: Figures 3, 4 and 5. The use of stars for the significant differences may be confused with the same sign used for identifying the treatments with a fixed initial total acidity of 10 meq/L

A2-09: For disambiguation the stars used to signal significant differences were replaced by “+” symbols. Figure captions were modified accordingly.

Discussion and conclusions

R2-10: Pay attention in these two sections to the inevitable influences of non-controlled unexpected factors, which are always randomly present in natural products. The comments are excessively detailed cycle by cycle. Focus on trends.

A2-10: The experiments were carried on in laboratory in strictly controlled conditions. The experimental design is rigorous and the effects observed are not random as demonstrated by the statistical analysis. Modification was done L74-L77:

“The aim of the study, carried on in laboratory conditions, is to assess the reproducibility of fermentation kinetics between backslopping cycles. The response of the microbial consortia was evaluated in terms of populations and activity (i.e. their modulation) to abiotic and biotic production parameters.”

Also, we believe that the description and interpretation must be appropriately detailed to guide the reader.

Comments on the Quality of English Language

R2-11: Some lenguage revision requiered.

A2-11 : Language was revised, namely L33-39 and L238-L241

Round 2

Reviewer 2 Report

Comments and Suggestions for Authors

The manuscript was improved